# Experimental identification of topological defects in 2D colloidal glass

Vinay Vaibhav [1,8], Arabinda Bera [1,8], Amelia C. Y. Liu [2],
Matteo Baggioli [3,4] ✉, Peter Keim [5,6,7] ✉ & Alessio Zaccone [1] ✉

Topological defects are singularities within a field that cannot be removed by continuous transformations. The definition of these irregularities requires an ordered reference configuration, calling into question whether they exist in disordered materials, such as glasses. However, recent work suggests that well-defined topological defects emerge in the dynamics of glasses, even if they are not evident in the static configuration. In this study, we reveal the presence of topological defects in the vibrational eigenspace of a two-dimensional experimental colloidal glass. These defects strongly correlate with the vibrational features and spatially correlate with each other and structural "soft spots", more prone to plastic flow. This work experimentally confirms the existence of topological defects in disordered systems revealing the complex interplay between topology, disorder, and dynamics.

Topological defects (TD) represent a ubiquitous hallmark of nature across different scales and they are generally defined as singularities in a local order parameter. They can appear in a wide range of physical systems[1–3], including but not limited to liquid crystals, superconductors, superfluids, ferromagnets, biological systems[4], and also our early universe[5]. Despite being microscopic in nature, TD can exert macroscopic influences on the behavior of the entire system. For example, they have a strong influence on the optical properties of liquid crystals and play crucial roles in various biological processes, including cell division, tissue formation, and the motion of cellular aggregates[6].

In the realm of condensed matter physics, the emergence of these defects disrupts ordered states[7] influencing collective excitations and even triggering phase transitions, such as the Kosterlitz, Thouless, Halperin, Nelson, Young (KTHNY) transition[8] in two-dimensional solids, investigated e.g. in a colloidal monolayer[9]. These defects possess distinct (topological) charges and exhibit robustness against continuous structural deformations, rendering them essential for understanding the fundamental properties of materials and being

totally different from other non-topological defects such as vacancies or interstitials.

Topological defects in 3D crystalline solids manifest as dislocations and disclination lines[7], providing structural irregularities within the otherwise ordered lattice, and the elementary carriers of plasticity. Dislocations, in particular, allow for the motion of crystalline planes sliding over each other (glide motion). The plasticity of crystalline solids is by now fully understood in terms of dislocation dynamics and dislocation networks, based on seminal concepts introduced by Taylor[10], Polanyi[11] and Orowan[12]. In 2D, topological defects are point defects and their thermal dissociation causes the elastic moduli to disappear[13].

In amorphous solids, instead, the absence of long-range order complicates the identification and characterization of topological defects (since, by definition, an ordered background is needed in order to detect its irregularities), posing a big challenge in linking structure with dynamics and predicting mechanical properties from the undeformed material configuration. The chase for structural topological defects in amorphous solids, borrowing from the concepts widely

[1]Department of Physics "A. Pontremoli", University of Milan, via Celoria 16, 20133 Milan, Italy. [2]School of Physics and Astronomy, Monash University, Clayton 3800 VIC, Australia. [3]Wilczek Quantum Center, School of Physics and Astronomy, Shanghai Jiao Tong University, Shanghai 200240, China. [4]Shanghai Research Center for Quantum Sciences, Shanghai 201315, China. [5]Institute for Experimental Physics of Condensed Matter, Heinrich-Heine-Universität Düsseldorf, 40225 Düsseldorf, Germany. [6]Max-Planck-Institute for Dynamics and Self-Organization, 37077 Göttingen, Germany. [7]Institute for the Dynamics of Complex Systems, University of Göttingen, 37077 Göttingen, Germany. [8]These authors contributed equally: Vinay Vaibhav, Arabinda Bera. ✉e-mail: b.matteo@sjtu.edu.cn; peter.keim@uni-duesseldorf.de; alessio.zaccone@unimi.it

used for crystals, is certainly not new and dates back to the early 70's[14]. Despite various efforts[15–26], it was nevertheless concluded that topological quantities cannot be properly defined by looking at the structural characteristics of glasses.

Therefore, apart from isolated cases[27,28], the identification of the plastic carriers and of the "soft spots", the regions (or particle clusters) more prone to plastic flow, in glasses remained based until now on the so-called structural indicators[29] and on the phenomenological concept of shear transformation zones (STZ)[30,31]. On the other hand, structural "soft spots", where particle rearrangements are initiated, have been postulated to be analogous to dislocations in crystalline solids[32] but lacking any formal definition and topological character.

Recent advances in simulation and theory[33] have pursued the idea of looking for topological defects in glasses using dynamical quantities such as the displacement vector field or the eigenvector field of normal mode vibrations, instead of the seemingly featureless structure. Following this idea, Baggioli et al.[34] have unveiled well-defined topological defects in polymer glasses under shear deformation. These topological defects are predicted by theory[35] as singularities in the microscopic displacement field, and are mathematically described by a continuous-valued Burgers vector in accord with an early intuition of Kleman and Friedel[36]. In amorphous solids, the atomic displacement field under an external deformation comprises two vector fields, one of which is fully ordered and is called the affine displacement field. This is simply the trajectory along which each atom moves as a consequence of the external strain field. The other vector field is known as the nonaffine displacement field and is much more random and irregular. It arises due to the local force imbalance on each atom in its affine position, where it is subject to a non-vanishing force (due to the lack of inversion symmetry) communicated by its neighbors[37]. In this sense, the affine component of the displacement field represents the ordered background disrupted by the topological defects living within (but not being equivalent to) the nonaffine displacements.

Following a similar logic, recent simulations of two-dimensional glassy model systems at $T = 0$ by Wu et al.[38] have revealed the presence of well-defined topological defects (of different nature) within the eigenvector field of vibrational modes, defined using the concepts of winding number and vortex structure[7]. In this second scenario, the ordered reference configuration, with respect to which defects are identified, is provided by the plane-wave eigenvector field. Both methods[34], and[38], have demonstrated a tight connection between these TD and plastic deformations, proving the capability of these structures to predict plastic spots and to correlate with the yielding transition.

Even more recently, mixing the ideas of[34] and[38], numerical investigations by Falk and collaborators[39] have identified saddle-like topological defects with quantized -1 charge in the displacement field, generating Eshelby-like quadrupolar fields which align to form the shear bands responsible for the yielding of amorphous solids. Interestingly, shear banding is directly related to the percolation of STZ, whose dynamics have been shown[40] to be intimately connected to vortex-like structures as those proposed by Wu et al.[38]. Finally, although topological charge neutrality is always enforced by the conservation of global elastic dipole charge, the yielding process in amorphous systems has been related to the clustering of net negative topological charge[41].

Topological defects in crystals can be also defined in terms of geometrical properties (curvature and torsion)[7]. A similar approach based on geometry has been recently proposed by Moshe, Procaccia and collaborators as the origin of plastic screening in amorphous materials[42], where the defects are identified with the elastic (dipolar and quadrupolar) charges. Interesting results in this context can be also obtained in terms of a tensorial form of electromagnetism known as vector charge tensor gauge theory that is able to reproduce the structure of static stress correlations in granular materials[43].

Importantly, all such characterisations of topological defects in amorphous materials have been performed using numerical simulations. Until now, no direct observation of topological defects in experimental amorphous systems has been reported. Given the importance of topological predictors of physical properties from biological tissues to cosmology, being able to detect topological defects in disordered structures experimentally is per se a fundamental goal of contemporary science.

Colloidal glasses have been proved to be an excellent experimental setup to test the validity of various measures and theories related to structure and dynamics[44]. Using optical microscopy, one has direct access to the structural information at the particle level. In this experimental study, we use bidisperse super-paramagnetic colloidal particles which are confined by gravity at an atomically flat interface to form a two-dimensional disordered structure. Applying an outer magnetic field perpendicular to the monolayer, the particles interact via well defined dipole-dipole repulsion. This gives a specific type of colloidal particles which become magnetised in the presence of a magnetic field and form a two-dimensional disordered structure. The particles interact via dipole-dipole repulsive interaction, so, based on particle positions, we can also directly infer the pairwise interaction energy. Using this information, we construct the dynamical matrix (Hessian) of the system, and perform the identification of topological defects in the field of normal modes. In this process, the analysis of the experimental data reveals unique features of vibrational characteristics of the system, and also the corresponding correlation with topology and "soft spots". Our experimental study demonstrates the existence of well-defined topological defects in the vibrational field of a finite temperature 2D colloidal glass, suggesting further experimental studies that can employ these defects to understand the thermal and mechanical properties of many complex, disordered systems.

## Results
### Experimental setup
Our experimental setup comprises a colloidal monolayer, where individual particles sediment to a flat water/air interface in a hanging-droplet configuration under the influence of gravity. These colloidal particles have two different masses and diameters, forming a binary mixture (Fig. 1b), and exhibit Brownian motion within the two-dimensional plane (see "Methods" and section 1 of the Supplementary Material for further details on experiment). The particles consist of polystyrene with incorporated nanograins of iron oxide, thus interactions are paramagnetic in nature, and a magnetic dipole moment is induced in each particle when an external magnetic field is applied perpendicular to the surface. The particles are sterically stabilized with Sodium Dodecyl Sulfate (SDS) slightly below critical micelle concentration. As anionic tenside, residual surface charges of the colloids are effectively shielded by counter-ions. The hard-core diameter (including the SDS layer) is never probed by $k_B T$ and the interaction is given by a pairwise dipole-dipole potential of the form[45,46] $E_{pot} \sim 1/r^3$, where $r$ is the inter-particle distance. The strength of the interaction can be controlled by the external magnetic field $H$, setting the characteristic energy scale of the system. A crucial parameter governing the system's behavior is represented by $\Gamma = E_{pot}/E_{kin}$, denoting the ratio of potential energy (stemming from this mutual dipolar interaction among particles) to the kinetic energy ($E_{kin} \sim T$) arising from thermal motion[47]. It can be interpreted as an inverse temperature or more didactically as dimensionless pressure. For our study, we set $\Gamma = 423$, a value at which the system is deep in a glassy state[47]. Optical microscopy is employed to record the particle positions in the field of view at a certain interval of time (Fig. 1a). The experimental setup is described in detail in[48], while the structure and dynamics of this system have been studied in previous work[45,46].

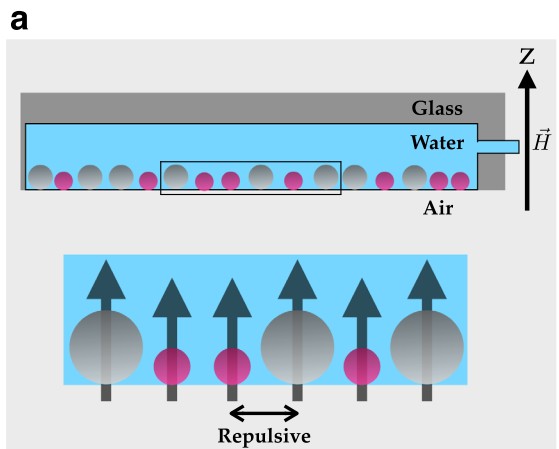

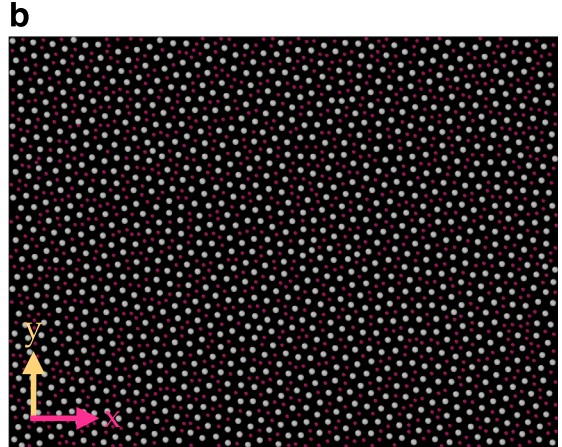

**Fig. 1 | Experimental realization of the colloidal monolayer forming a glass.**
**a** Schematic showcasing the binary colloidal mixture with large and small particles colored in silver and pink (side view, not to scale). A water droplet hanging by surface tension inside a cylindrical hole in a glass plate, hosts the mixture of colloidal particles at the bottom water-air interface. The volume of water droplet is

actively regulated via a computer-controlled microsyringe to ensure a flat interface. The zoomed portion shows the dipole moment induced in each colloidal particle in the presence of external magnetic field $\vec{H}$ perpendicular to the monolayer that leads to repulsive dipolar interaction. **b** Top view of the arrangement of colloidal particles, reconstructed from positional data by video microscopy.

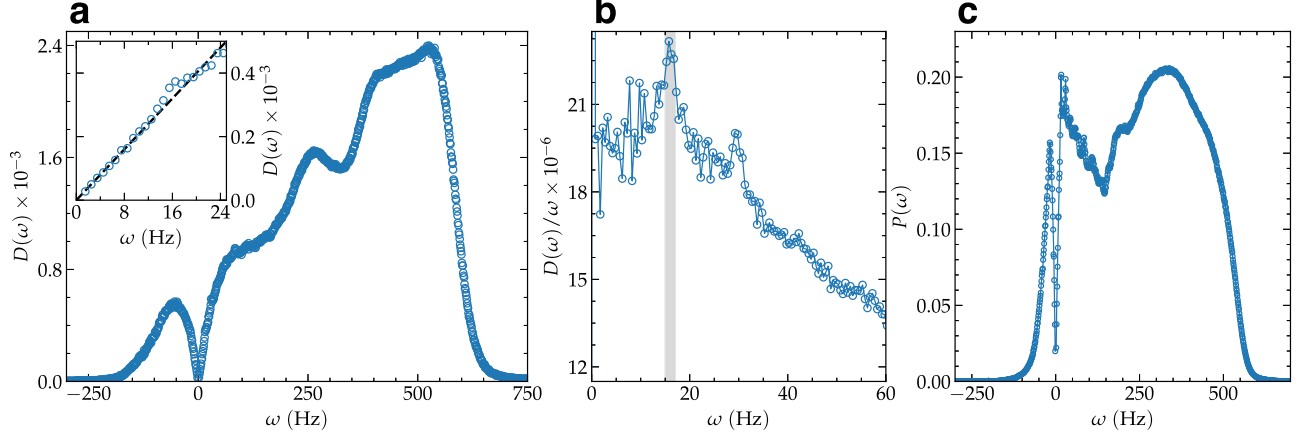

**Fig. 2 | Vibrational characteristics of the system. a** The vibrational density of states $D(\omega)$ is presented, with the inset highlighting the low-frequency behavior. The dashed line represents a linear trend, consistent with Debye's prediction $D(\omega) \sim \omega^{d-1}$, where $d = 2$. **b** $D(\omega)/\omega$ versus $\omega$ is plotted at low frequencies, with shaded

regions indicating the occurrence of the boson peak. **c** The participation ratio $P(\omega)$ versus $\omega$ is depicted, providing insights into the localization properties of vibrational modes across different frequency ranges.

## Vibrational properties

Using the long-range dipole-dipole interaction potential among the particles, we numerically construct the dynamical (Hessian) matrix which is diagonalized to obtain the eigenvectors $\vec{e}_l$ and eigenfrequencies $\lambda_l$. Here, $\lambda_l$ indicates the l-th eigenvalue with $l = 1, ..., 2N$; $N$ being the number of particles in the field of view. From the eigenvalues, we derive the corresponding eigenfrequencies $\omega_l^2 = \lambda_l$ (using a mass-rescaled Hessian, see "Methods"). As expected from amorphous solids at finite temperature[49], the spectrum displays a fraction of negative eigenvalues, corresponding to unstable modes with purely imaginary frequency. We notice that the presence of unstable modes is not incompatible with the solid nature of our experimental system. In fact, unstable modes exist in glasses[49] and even heated crystals[50], as a result of local anharmonicities. More in general, these unstable modes correspond to dynamics over regions of the potential landscape with locally negative curvature e.g., saddle points and potential barriers and they are widely observed in supercooled liquids as well[51]. For the unstable part of the spectrum ($\lambda < 0$), we follow the practice of re-defining a positive definite frequency $\tilde{\omega} = -i\sqrt{\lambda}$ and plotting the corresponding vDOS on the negative frequency axes upon identifying

$\omega = -\tilde{\omega}$. The results for the experimental vibrational density of states (vDOS) $D(\omega)$ are shown in Fig. 2a.

As highlighted in the inset of Fig. 2a, the low frequency behavior of the vDOS shows a linear scaling, $D(\omega) \sim \omega$. This behavior is compatible with Debye's law in two spatial dimensions but it is also a characteristic feature of the vDOS of systems with unstable modes[52,53]. Interestingly, we observe that the vDOS is symmetric at low frequency and the linear scaling (including the corresponding slope) is the same for the stable and unstable branches. We also notice that only the stable branch can be directly interpreted as pertaining to vibrational modes stricto sensu. We have further tested the robustness of these findings by introducing a cut-off distance $R_c$ in the interaction potential, incorporating random polydispersity in particle masses, and accounting for experimental measurement errors in distances (see Supplementary Figs. 3 and 4). Our results demonstrate that the characteristics of the vDOS remain robust against variations in $R_c$, polydispersity, and possible experimental measurement errors.

In Fig. 2b, we present the same vDOS normalized by the low-frequency linear (Debye) scaling, $D(\omega)/\omega$. In this representation, the presence of an anomalous peak is clear around a characteristic

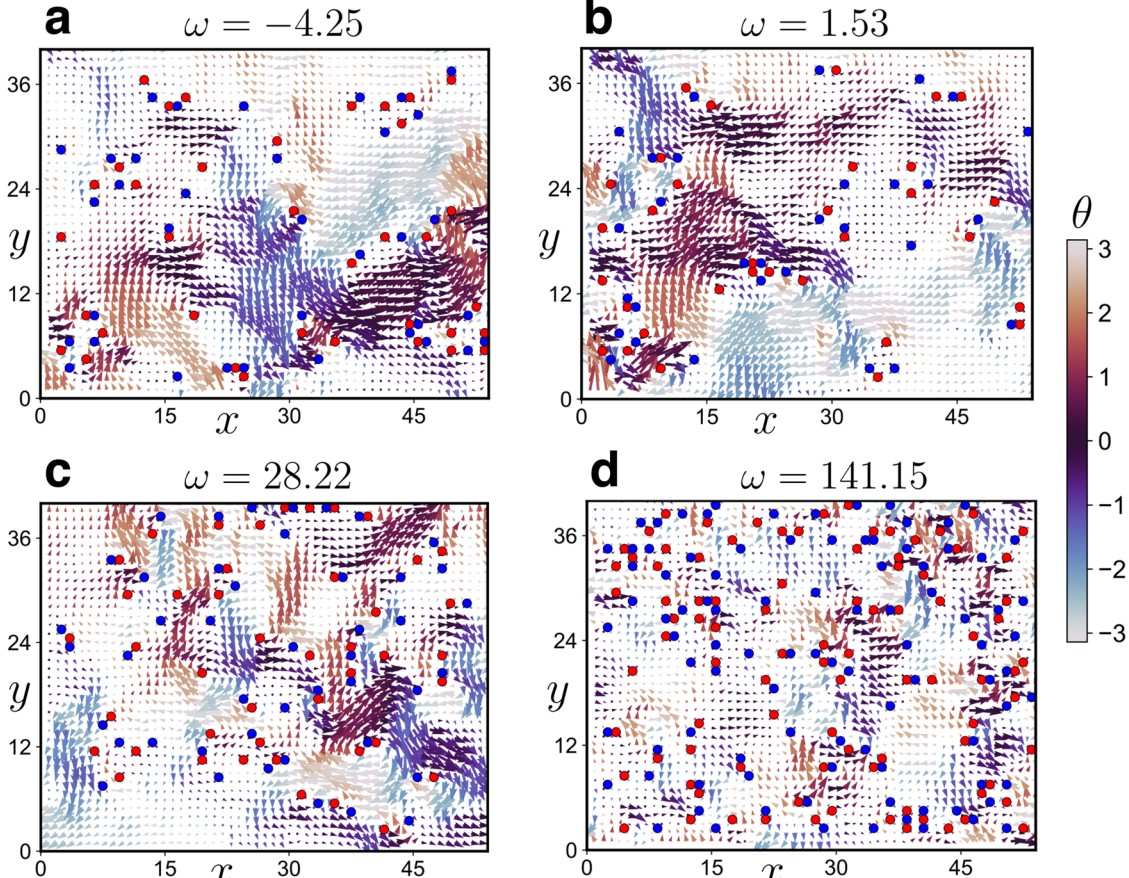

**Fig. 3 | Topological defects in the eigenvector field of the glass. a–d** Eigenvector fields within a rectangular field of view ($54a \times 40a$, where $a = 20.78$ μm) are depicted for various eigenfrequencies ($\omega = -4.25$, 1.53, 28.22, and 141.15). The color bar denotes the phase of the eigenvectors ($\theta = \tan^{-1}(e_y/e_x)$), capturing the directional information of the vibrational modes. The size of the arrowhead indicates the magnitude of the eigenvector field at that point. Topological defects are indicated by filled circles, with red and blue representing $q = +1$ and $-1$ respectively, highlighting the presence of non-trivial topological features within the two-dimensional colloidal glass system.

frequency $\omega_{BP}$, marked within error bars by the shaded vertical strip (15–17 Hz). This vibrational anomaly over Debye's law is known as "boson peak" and it is a common feature of amorphous materials[54]. Despite the huge effort devoted to characterize and comprehend this feature in 3D systems, its experimental detection in quasi two-dimensional (2D) amorphous materials remains limited[55,56]. In this respect, our result provides another experimental identification of the BP anomaly in a specific type of 2D amorphous materials—colloidal glasses—using directly the Hessian matrix rather than the covariance matrix[57]. In these colloidal systems, the density of states using covariance matrix have been extensively studied previously[47,57]. In anharmonic systems like ours, the vDOS obtained from the covariance matrix formalism differs significantly from that obtained through normal mode analysis (see, e.g.,[58]). Therefore, a direct comparison between these two methods appears not trivial and requires further systematic investigations. Also, the unit of $\omega$ (derived from the normal mode analysis) is sec$^{-1}$, while the frequency unit obtained from the covariance matrix is $\mu m^{-1}$ [47,57], making this comparison more difficult.

To measure the extent of mode localization, we also calculate the participation ratio $P(\omega)$, plotted in Fig. 2c. We observe a sharp peak corresponding to acoustic phonon-type excitations at very low (positive) frequency, and a secondary broader peak at a higher frequency possibly linked to optical-like excitations due to the particle size mismatch. This secondary peak then drops sharply in correspondence of the Anderson-localized highest frequency modes which terminate at the Debye frequency of the system.

## Topological defects

We move to the analysis of the spatial structure of the vibrational modes by investigating the corresponding eigenvector fields. In order to identify and characterize the topological properties of the eigenvector field, we resort to the method originally proposed by Wu et al.[38], based on the computation of the local winding number $q$. By considering the smallest square grid in our experimental data, we identify vortices (anti-vortices) as topological defects with winding number (or equivalently topological charge) $q = +1$ ($-1$).

Figure 3a–d provide visual representations of the eigenvector fields at various frequencies, with filled red and blue circles denoting the locations of vortices and anti-vortices, respectively. Additionally, the color map indicates the local angle of the eigenvector field with respect to the $x$-axes. The structure of the eigenvector field is evidently highly heterogeneous, displaying several vortex-like structures and singularities. Notably, the eigenvectors at lower frequencies exhibit a smaller number of topological defects. This suggests that the number of defects grows with frequency and becomes more uniformly distributed in space. Significantly, at lower frequencies, there are fewer topological defects, and a rich, cooperative, and swirling eigenfield structure. This might be related to the fact that at low-frequency the dynamics are dominated by plane waves, displaying a periodic structure of swirling (see e.g. Fig. 3 in[59]). Conversely, at higher frequencies, there is a notable increase in defect density, with defects uniformly distributed throughout the space and reduced coherence in the eigenvector field. Interestingly, topological defects can be also found

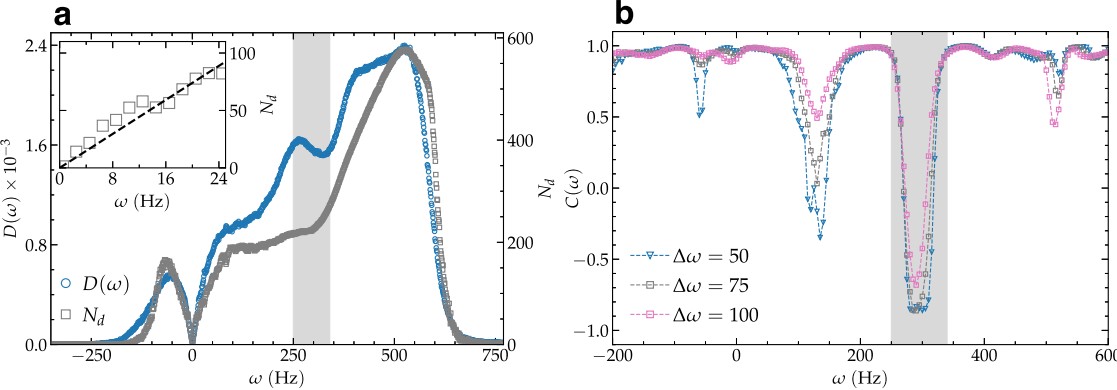

**Fig. 4 | Correlation between vibrational density of states and number of topological defects. a** The vibrational density of states ($D(\omega)$) and the total number of defects ($N_d$) are plotted against vibrational frequencies. The left-side vertical axis corresponds to $D(\omega)$, while the right-side axis corresponds to $N_d$. The inset illustrates the low-frequency behavior, with dashed line indicating linear trend.

**b** Pearson correlation coefficients ($C$) between $D(\omega)$ and $N_d(\omega)$ are depicted for varying $\omega$, considering three different frequency widths: $\Delta\omega = 50$, 75, and 100. The shaded gray area marks the frequency range $\omega \in [250, 340]$, where strong anti-correlation is observed in both panels (**a, b**).

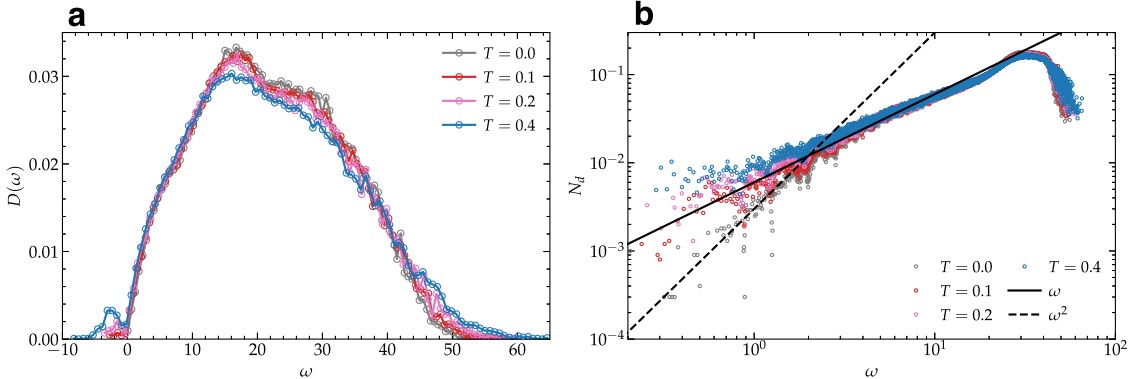

**Fig. 5 | Simulation of 2D Lennard-Jonnes glass[38] at finite temperature. a** Vibrational density of states $D(\omega)$ and (**b**) number of topological defects $N_d$ as a function of frequency $\omega$ are shown in linear and double-log scale, respectively, at different temperatures for a two dimensional Lennard-Jones glass.

in the eigenvectors corresponding to unstable modes, e.g. panel a in Fig. 3.

Figure 4a presents the vibrational density of states $D(\omega)$ and the total number of defects $N_d$ against vibrational frequencies within the same frame. This highlights a robust correlation between these quantities across the entire frequency spectrum. The inset of Fig. 4a delves into the low-frequency behavior of $N_d$, where the dashed line signifies the linear dependence on $\omega$. To further establish the correlation between vibrational properties and topological defects, we calculate the Pearson correlation coefficient, $C(\omega)$, at various $\omega$ values, considering data points within a specified frequency range $\omega - \Delta\omega/2$ to $\omega + \Delta\omega/2$ (see "Methods"). Figure 4b illustrates the variation of $C(\omega)$ for different frequency width values ($\Delta\omega = 50$, 75, and 100). We observe that the fluctuations of $C(\omega)$ decrease as the frequency width increases. The proximity of $C$ values to unity across the majority of the spectrum signifies a robust correlation between $D(\omega)$ and $N_d$. However, intriguingly, instances of strong anti-correlation are observed within specific frequency ranges, as highlighted by gray vertical stripes in Fig. 4a, b. Particularly noteworthy is the presence of such anti-correlation within the frequency range $\omega \in [250, 340]$, prompting further theoretical investigation to unravel its origin and its consequential impact on the system's structure and dynamics. This observed anti-correlation arises because, in this frequency range, the DOS is decreasing while the number of defects continues to increase gradually. At this moment, we do not have a concrete understanding of the physical significance of these anti-correlations appearing at large frequencies, above the

Debye regime where both $N_d(\omega)$ and $D(\omega)$ scale like $\omega$ (and nicely correlate).

It is important to notice that our scaling $D(\omega) \propto \omega$ at low frequency seems at first sight in contradiction with the results reported in ref. 38. In order to reveal the cause of this discrepancy, we emphasize that the results of ref. 38 are obtained for a zero temperature system. Here, we perform additional simulations in a 2D Lennard-Jones glass at zero as well as finite temperatures (see Section 6 of the Supplementary Material). At $T = 0$, we reproduce the results in ref. 38, where $N_d$ scales quadratically with the frequency. We further extend our analysis at finite temperatures. As expected, the fraction of unstable modes grows with temperature, in contrast to the athermal case where such modes are absent, as shown in 5a. Our focus is on the small $\omega$ behavior of $N_d$ with the temperature change. In Fig. 5b we observe that, unlike for the athermal case where $N_d \sim \omega^\alpha$ with $\alpha = 2$, the scaling of the number of defects with frequency becomes linear at finite temperature. The scaling of the number of defects in the experimental system shows an exponent $\alpha$ close to 1, compatible with the simulation results at finite $T$. In Fig. 5b, we show that this difference is due to the finite temperature dissipative effects and we recover the results of [38] in the zero temperature limit, solving the apparent contradiction.

## Structural features of defects and their correlation with soft spots

To understand the spatial organization of defects, we calculate partial pair correlation functions $g_{PP}(r)$, $g_{PN}(r)$ and $g_{NN}(r)$ for positive-positive,

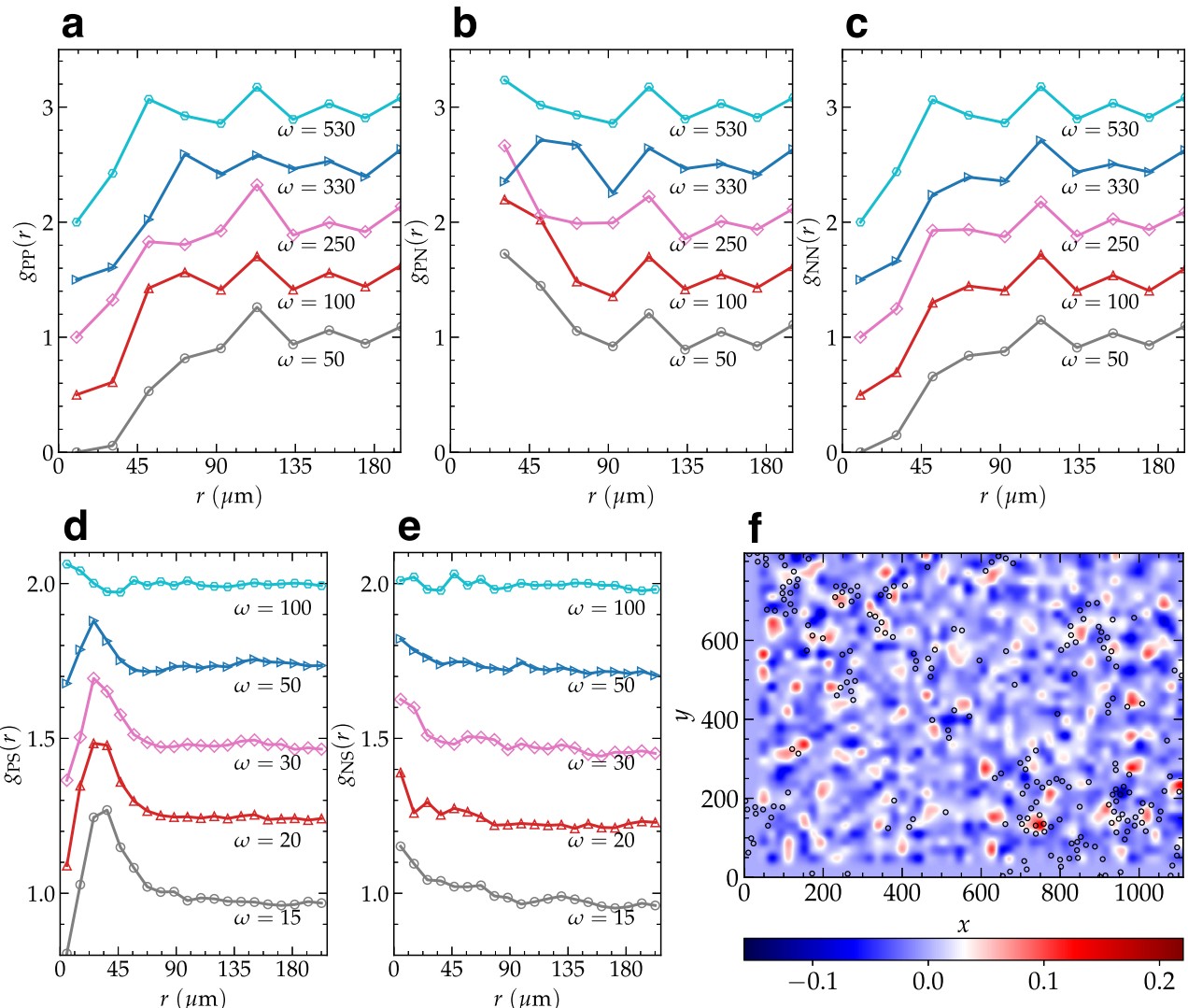

**Fig. 6 | Structure of defect pairs and the spatial correlation between defects and soft spots. a–c** Pair correlation functions for positive-positive, positive-negative and negative-negative defect pairs ($g_{PP}(r)$, $g_{PN}(r)$ and $g_{NN}(r)$), respectively, for different $\omega$ values. Plots other than $\omega = 50$ are shifted by the multiple of 0.5 for clarity. **d, e** Spatial correlation between soft spots (with top 10% softness) and topological defects with, **d,** positive charge $g_{PS}(r)$, and, **e,** negative charge $g_{NS}(r)$, for different $\omega$ values. For better clarity, curves other than $\omega = 15$ are shifted upward by the multiple of 0.25. **f** Color map of charge density field obtained by the $\omega^{-1}$ weighted averaging of topological charge over frequencies in the range $0 < \omega < 50$, shown together with the soft spots (open symbols).

positive-negative and negative-negative defect pairs respectively, as shown in Fig. 6 (see "Methods" for definition). From the positive-positive and negative-negative defect correlation in Fig. 6a, c, it is clear that, for small $r$, there is no correlation (or even a "depletion" zone near contact), indicating the absence of defects with similar charge in the vicinity of a tagged defect. Instead, the positive-negative correlation in Fig. 6b shows a very high probability for the presence of another defect with an opposite charge in the neighborhood of a tagged defect. This observation suggests that defect charges with the same sign repel each other and those with opposite sign attract each other. It appears that the repulsion between two defects both with negative sign is slightly stronger because the repulsive "hard-core" distance at small separation $r$ in $g_{NN}$ is bigger than that in $g_{PP}$. Overall, both attraction and repulsion become weaker with the increase in frequency $\omega$, and at sufficiently high $\omega$ such defects are expected to be completely uncorrelated and homogeneously distributed.

In recent studies, it has been possible to identify "soft spots" in the low-frequency vibrational modes where mesoscale relaxation or rear-rangements are prone to happen[32,60]. We identify such soft spots in the present system using the softness field defined in[60] and calculate their radial pair correlation with positive ($g_{PS}(r)$) and negative ($g_{NS}(r)$) defects (see "Methods" and Supplementary Fig. 7). Figure 6d–e shows the correlation for $\omega = 15, 20, 30, 50, 100$. It is quite evident that the defects with negative charge are highly correlated with soft spots at small $r$, while defects with positive charge show no such correlation. This means negative charge defects tend to appear in the close vicinity of soft spots. Furthermore, it will be expected that these defects with negative charge will most likely be associated with mobile regions in structural relaxation or plastic rearrangements under deformation. Defects with positive charge located somewhat further away from the soft spots are also correlated, evidenced by the peak in $g_{PS}$ at $r \approx 40$ μm. This is expected because we have already seen a significant correlation between +1 and -1 defects at the same length-scale, see Fig. 6b. With the increase in $\omega$, the correlation becomes weaker with both types of defects. Fig. 6f shows soft spots (open sysmbols) with the color map of weighted charge density field obtained by the averaging of topological charge in the frequency range $0 < \omega < 50$ (see Methods for the definition). This visual representation demonstrates the correlation between the zones with high negative charge density and soft spots.

## Discussion

Based on our experimental findings and analysis, we have successfully explored the topological characteristics of a two-dimensional colloidal glass system under the influence of an external magnetic field. Through meticulous control of experimental parameters and numerical techniques, we have revealed intriguing insights into the interplay between topology, vibrational properties, and defect dynamics within disordered systems.

Our investigation has unveiled the presence of topological defects within the eigenspace of vibrational frequencies of an experimental 2D colloidal glass. This provides a direct experimental observation of topologically non-trivial features in a colloidal glass. Notably, the observed correlation between the vibrational density of states (vDOS) and the total number of defects underscores the profound influence of topology on the material's vibrational behavior. The robustness of this correlation, as evidenced by the Pearson correlation coefficient analysis, highlights the significance of topological defects in shaping the structural and dynamical properties of colloidal glasses.

By analyzing the structural properties of the topological defects, we have shown that defects of opposite charge pair together while defects with same charge repel each other, as expected from our intuition from electromagnetism. More importantly, we have observed a strong correlation at short distance between defects with negative topological charge and soft spots, confirming a close relation between anti-vortices and plasticity as suggested in[38]. Interestingly, we have found that a correlation between positive defects and plastic spots exists at larger length-scales as well.

Our experimental analysis complements and confirms several of the results previously obtained from simulations at zero temperature[38], but it also provides several important new lessons and clarifications. First, it proves that the topological vortex-like defects proposed in[38] survive at finite temperatures. Second, it clarifies that the number of these defects in realistic, and necessarily finite temperature, systems does not scale quadratically with frequency, as argued in[38], but rather linearly with it, nicely correlating with the vibrational properties of the system. Finally, our experimental data supports the claim of[38] of a strong local correlation between the −1 defects and plastic spots at short scales. Nevertheless, they also reveal a strong correlation with the +1 defects at an intermediate larger scale, that is consistent, and indeed expected, from the strong correlation among +1 and −1 defects. This suggests that +1 and −1 defects tend to pair together and create dipole structures, with possible intriguing connections with recent results about plastic screening in amorphous systems by Lemaître et al.[42].

In conclusion, our study on a sedimented 2D colloidal glass with ideal dipole-dipole interactions demonstrates the existence of mathematically well-defined topological defects in a completely disordered experimental system. This finding suggests that similar analytical approaches may be usefully extended to other structural glasses. It contributes to advancing our understanding of the intricate interplay between topology, vibrational properties, and defect dynamics in disordered systems. The insights gained from this research not only deepen our knowledge of disordered materials, but also pave the way for future explorations aimed at uncovering the fundamental principles governing the mechanical and thermal behavior of complex materials. As our understanding continues to evolve, we anticipate that further studies will shed light on new phenomena related to the present observations, and unveil novel avenues for exploration in condensed matter physics, biology, materials science and cosmology.

## Methods
### Experimental details
We consider here an experimental setup of an equimolar binary colloidal mixture (consisting of species A and B) of spherical particles in 2D, with two different magnetic moments. The two species have diameters $d_A = 4.5$ μm, $d_B = 2.8$ μm, magnetic susceptibilities (per particle) $\chi_A = 6.2 \times 10^{-11}$ Am$^2$/T, $\chi_B = 6.6 \times 10^{-12}$ Am$^2$/T and mass densities $\rho_A = 1.5$ g/cm$^3$, $\rho_B = 1.3$ g/cm$^3$. A constant magnetic field $H = 3.9 \times 10^{-3}$ T is applied perpendicular to the plane containing the particles which induces magnetic moment $M_A = \chi_A H$ or $M_B = \chi_B H$ in each particle, thus, particles interact via the dipole-dipole pair potential with each other. The potential energy between two constituent particles separated by a distance $r$ is given by

$$V_{\alpha\beta}(r) = \frac{\mu_0}{4\pi} \frac{M_\alpha M_\beta}{r^3}, \tag{1}$$

with $\alpha, \beta \in \{A, B\}$ and $\mu_0 = 4\pi \times 10^{-7}$ Tm/A is the vacuum permeability. Here we have approximately $N = 2300$ number of particles in the rectangular field of view 1158 × 865 μm$^2$ in each sample.

### Vibrational analysis
We obtain eigenvalues $\lambda_l$ ($l = 1, 2, ..., 2N$) and associated eigenmodes by diagonalizing the dynamical (Hessian) matrix given by

$$H_{ij} = \frac{1}{\sqrt{m_i m_j}} \frac{\partial^2 U}{\partial r_i \partial r_j}. \tag{2}$$

Here $m_i$ and $r_i$ are the mass and spatial coordinates of the $i$th colloidal particle respectively. Also, $U = \sum_{\alpha < \beta} V_{\alpha\beta}$ is the potential energy of the system. Further, we estimate the eigenfrequencies as $\omega_l = \sqrt{\lambda_l}$. From the distribution of $\omega$, we obtain the vibrational density of states (vDOS)

$$D(\omega) = \frac{1}{2N-2} \sum_l \delta(\omega - \omega_l). \tag{3}$$

Note that, we conventionally represent the imaginary frequencies with negative values while showing the density of states.

### Characterization of topological defects
For each eigenvector field $(e_i^x, e_i^y)$ at $\omega_i$ ($i = 1, 2, ..., 2N$), we assign an angle $\theta(\vec{r})$ on every site at $\vec{r}$ of a 54 × 40 rectangular lattice having grid length $r_c$ in both the directions and superposed to the experimental system. We obtain the phase angle $\theta(\vec{r})$ at each lattice site as[38]

$$\tan\theta(\vec{r}) = \frac{\sum_i w(\vec{r} - \vec{r}_i) e_i^y}{\sum_i w(\vec{r} - \vec{r}_i) e_i^x}, \tag{4}$$

where $\vec{r}_i$ is the location of particle $i$, and $w(\vec{r} - \vec{r}_i)$ is a Gaussian weight function, defined to be $w(\vec{r} - \vec{r}_i) = \exp(-|\vec{r} - \vec{r}_i|^2/r_c^2)$ with $r_c = 20.78$ μm (lattice spacing).

We determine the topological charge $q$ inside each smallest square grid by evaluating the line integral of $\vec{\nabla}\theta$ over a closed path inside the lattice, given by

$$q = \frac{1}{2\pi} \oint \vec{\nabla}\theta \cdot \vec{d\ell}, \tag{5}$$

where $\vec{d\ell}$ represents the line element along the closed square loop. Typically, the value of $q$ is an integer, and it is used to identify the locations of defects, which are characterized by non-zero values of $q$ at the center of the smallest square regions. For the calculation of topological charge, we do not enforce the periodic boundary conditions. We also show a low-frequency configuration at $\omega = 1.53$ Hz for three different interpolation grid lengths ($r_c$) in Supplementary Fig. 5, to show the robustness of the defect identification against the choice of grid lengths[61]. Further, these configurations are presented using Schlieren patterns[62,63], where defect locations are marked by the merging of distinct color intensities, while uniform color regions indicate defect-free areas (see Supplementary Figs. 5d±f).

Such patterns are consistent with the visualization of defects identified using eq. (5).

We calculate the total number of defects $N_d$ by counting the defects with $q = +1$ and $-1$. This defect behavior remains robust against slight variations in the interpolation grid length $r_c$ (see Supplementary Fig. 6). For a given mode of eigenfrequency $\omega$, the radial pair correlation between topological defects $\alpha$ and $\beta$ with $\{\alpha, \beta\} \in \{P, N\}$, is defined as

$$g_{\alpha\beta}(r) = \frac{L_x L_y}{2\pi r N_\alpha N_\beta} \sum_{i=1}^{N_\alpha} \sum_{j=1}^{N_\beta} \delta(r - |\vec{r_{ij}}|). \qquad (6)$$

Here, $L_x$, $L_y$ are the dimensions of the region in which correlation is calculated, {P, N} are the positive and negative topological defects, $N_\alpha$ and $N_\beta$ are the respective numbers of such defects, and $|\vec{r_{ij}}|$ is the distance between defect $i$ and $j$.

### Calculation of correlation coefficients

The Pearson correlation coefficient ($C(\omega)$) is calculated as[64]

$$C(\omega) = \frac{\sum_{\{\omega\}} (D(\omega) - \bar{D}(\omega))(N_d(\omega) - \bar{N}_d(\omega))}{\sqrt{\sum_{\{\omega\}} (D(\omega) - \bar{D}(\omega))^2} \sqrt{\sum_{\{\omega\}} (N_d(\omega) - \bar{N}_d(\omega))^2}}, \qquad (7)$$

where $\bar{D}(\omega)$ and $\bar{N}_d(\omega)$ are the average value of $D(\omega)$ and $N_d(\omega)$, respectively, within the considered range of $\omega \in [\omega - \Delta\omega/2, \omega + \Delta\omega/2]$, where $\Delta\omega$ defines the frequency width.

To ensure a thorough examination of the correlation between $D(\omega)$ and $N_d(\omega)$ across the frequency spectrum, we employ the following approach. Initially, we compute the average data for $D(\omega)$ and $N_d$ across the entire range of $\omega$. Subsequently, we construct an interpolation function, enabling us to access the values of $D(\omega)$ and $N_d$ for any given $\omega$.

For each $\omega$, we interpolate $2 \times 10^4$ data points with increasing $\omega$ within the range $\omega \in [\omega - \Delta\omega/2, \omega + \Delta\omega/2]$. Utilizing this interpolated data, we then calculate $C(\omega)$ using Eq. (7). This meticulous procedure guarantees a comprehensive assessment of the correlation between $D(\omega)$ and $N_d(\omega)$, ensuring robust and reliable results across the entire frequency spectrum.

### Definition of soft spots and their correlation with topological defects

We calculate the softness field $\phi_i$ for each particle $i$ by superposing the participation fraction weighted by the corresponding mode energy in the low-frequency vibrational modes[32,60],

$$\phi_i = \frac{1}{N_m} \sum_{j=1}^{N_m} \frac{|\mathbf{e}_j^{(i)}|^2}{m_i \omega_j^2}. \qquad (8)$$

Here $\mathbf{e}_j^{(i)}$ is the eigenvector corresponding to particle $i$ associated with frequency $\omega_j \in (0, 50)$ and $N_m$ is the number of low energy modes within this frequency range. Further, we identify the particles with top 10% softness value as soft spots.

The radial pair correlation function between the soft spots and the topological defects with positive charge in mode $l$ is defined as

$$g_{PS}^l(r) = \frac{L_x L_y}{2\pi r N_d^l N_S} \sum_{i=1}^{N_d^l} \sum_{j=1}^{N_S} \delta(r - |\vec{r_{ij}}|), \qquad (9)$$

where $L_x$, $L_y$ are the dimensions of the region in which correlation is calculated, $N_d^l$ is the number of defects with positive charge in mode $l$ inside the region, $N_S$ is the number of soft spots inside the region and $|\vec{r_{ij}}|$ is the distance between topological defect $i$ and soft spot $j$.

Similarly, $g_{NS}^l$, correlation between defects with negative charge corresponding to $\omega_l$ and soft spot can be defined.

### Calculation of charge density

To calculate the topological charge density, we divided the experimental box into square cells of size $a_s = 20.78$ μm. Within each cell, we counted the number of topological defects $n_p$ and $n_n$ with winding numbers +1 and −1, respectively, for $\omega_i \in (0, 50)$. The average charge density within each square cell is defined as

$$\bar{c} = \frac{\sum_i (n_p - n_n)/\omega_i}{\sum_i 1/\omega_i}. \qquad (10)$$

Here, we employed the weighting factor $1/\omega_i$ to account for the linear dependence of the total number of defects on lower frequencies.

## Data availability

The data that support the findings of this study are available within the article and its Supplementary Information or from the corresponding authors upon request. The processed microscopy data used in this study is available at a dedicated GitHub repository (https://github.com/vinayphys/2DColloidTopoDefect)[65]. Source data are provided with this paper.

## Code availability

The codes that support the findings of this study are available upon request by contacting the corresponding authors.

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

## Acknowledgements

We would like to thank Yuliang Jin, Deng Pan, Piotr Surowka, Jie Zhang, Cunyuan Jiang, Yujie Wang, Zihan Zheng, Jack Douglas, Tim W. Sirk, and Chengran Du for many discussions on topological defects in glasses and related collaborations. M.B. acknowledges the support of the Shanghai Municipal Science and Technology Major Project (Grant No. 2019SHZDZX01) and the sponsorship from the Yangyang Development Fund. A.C.Y.L. acknowledges the support of the Australian Research Council (FT180100594). P.K. acknowledges funding of the German Research Foundation within the Heisenberg-

Program, project number 453041792. A.Z. and V.V. gratefully acknowledge funding from the European Union through Horizon Europe ERC Grant number: 101043968 "Multimech". A.Z. gratefully acknowledges the Niedersächsische Akademie der Wissenschaften zu Göttingen in the frame of the Gauss Professorship program. A.Z. and A.B. gratefully acknowledge funding from US Army Research Office through contract nr. W911NF-22-2-0256.

## Author contributions

P.K. provided the dataset of the experiment. V.V. and A.B. performed the numerical analysis. A.C.Y.L. assisted in the topological defect identification. A.Z. and M.B. supervised the project. All authors contributed to the writing of the manuscript.

## Funding

## Competing interests

The authors declare no competing interests.
