## [Transparent Peer Review file · Nature Communications]

Experimental identification of topological defects in 2D colloidal glass

Corresponding Author: Professor Alessio Zaccone

Version 0:

Reviewer comments:

Reviewer #1

(Remarks to the Author)

The authors well answered my questions in detail and the necessary revisions are added to the manuscript. I think that it can be published in Nat. Commun.

Minor point:

It is better to explicitly show the definition of correlations in Fig.6.
All reference numbers in the main text are "?" which are not properly shown.

Reviewer #3

(Remarks to the Author)

This manuscript was transferred from Nat.Phys. to Nat.Comm. after being reviewed by three reviewers. All reviewers acknowledged the originality and ambition of the research, particularly in defining, identifying, and characterizing new types of topological defects in disordered solids, and the attempt to provide experimental validation.

However, none of the reviewers recommended publication due to a significant weakness in the manuscript. In short, all reviewers felt that the experimental content of the work was incremental and that the model-based definition of defects was not sufficiently validated.

Each reviewer proposed additional analyses to validate different aspects of the defect definition or the interpretation of the experimental measurements. The authors chose not to implement most of these suggestions. Instead, they appropriately addressed the reviewers' concerns directly, rendering the suggested analyses unnecessary. Accordingly, the authors revised the manuscript and significantly strengthen it.

I believe the paper, in its current form, should be accepted for publication.

Minor comment:

The response to reviewer #1's question, "Why choose $\omega=50$ value?" is not entirely convincing. Based on the authors' reply, I believe they should have provided a clearer explanation of the parameters in the system that determine this value.

Reviewer #4

(Remarks to the Author)

As I understand it, the present work presented in this paper is a hybrid of experiment and numerics: an experimental colloidal glasses actual configuration is then simulated to compute a vibration spectrum. The idea is to experimentally confirm that some elements of the soft spots in the colloidal glass are correlated to the "defect" distribution of the vibration

eigenmodes of lowest frequency. The key paper to compare to are simulations in Ref. 38, which show a correlation between plastic events and topological defects.

The paper has obviously been reviewed by many experts already. I don't want to torture the authors too much on what is clearly an honest and reasonable work. I do have questions though that I would be interested seeing addressed.

One key difference between the figures of Ref. 38 and those of the present work that are bothering me are that the experimental eigenmodes seem to have large swaths of small magnitude vectors away from the defects (how small is not clear though so perhaps I am seeing an artifact of the graphics). That is, the regions between defects appear to be much less ordered here than in Ref. 38, and many of the defects identified appear in these low order regions (I am abusing the analogy to stat mech here a little). It makes intuitive sense that anti vortices would signify places that rearrangements might happen but surely the magnitude of the vibration around that defect must be important there? Might the authors at least comment on the apparent differences in how the eigenmodes actually look?

There are also other ways (maybe better ways) of visualizing topological defects beyond these discrete winding numbers (see Hoffmann, Karl B., and Ivo F. Sbalzarini. "Robustness of topological defects in discrete domains." *Physical Review E* 103, no. 1 (2021): 012602 on identifying defects in discrete systems). I think it might be useful to at least check that your identification of the defects are robust to other methods of identifying topological defects. A method I especially like is to look at the curves that are spanned by all the arrows pointing near a particular direction (maybe colored by intensity). This would be like looking at a Schlieren texture in a 2D nematic liquid crystal. I believe this is called the Pontryagin-Thom construction. These lines terminate on or near the defect cores (ideally) because defects are characterized by the vector order pointing in all directions as you traverse the core. See, for example, Alexander, Gareth P. "Topology in liquid crystal phases." *The role of topology in materials* (2018): 229-257.

Version 1:

Reviewer comments:

Reviewer #4

(Remarks to the Author)

I am very happy with the changes made by the authors and recommend publication.

Reply to Referees for Manuscript NCOMMS-24-55036-T

Dear Referees,

Thanks a lot for sending the second round of review for our manuscript. Here we address all your comments point by point. We also revised the manuscript based on your feedback and the changes are marked in **red color**. We also have updated some references with their latest versions.

We hope that the extensively revised version of our manuscript is now suitable for publication in *Nature Communications*.

Sincerely,
The Authors

Referee 1

The authors well answered my questions in detail and the necessary revisions are added to the manuscript. I think that it can be published in Nat. Commun.

We are happy to note that our response to the comments and revisions in the manuscript could satisfy the expectations of Referee 1. We thank him/her for recommending our work for publication in Nature Communications.

Minor point: It is better to explicitly show the definition of correlations in Fig.6. All reference numbers in the main text are "?" which are not properly shown.

In Fig.6 we have two different types of correlations. First is the pair-correlations ($g_{PP}(r), g_{PN}(r), g_{NN}(r)$) between topological charges to understand their spatial organization. The second is the pair correlations between topological charges and soft spots ($g_{PS}(r), g_{NS}(r)$), which have already been defined in Methods (see eq.(9) in the revised manuscript). We thank the referee for pointing out the missing definition of first type of correlations. Now, we have defined these correlations in Methods as eq.(6). The problem related to references might be due to some compilation issue; we have fixed it.

Referee 3

This manuscript was transferred from Nat.Phys. to Nat.Comm. after being reviewed by three reviewers. All reviewers acknowledged the originality and ambition of the research, particularly in defining, identifying, and characterizing new types of topological defects in disordered solids, and the attempt to provide experimental validation.

However, none of the reviewers recommended publication due to a significant weakness in the manuscript. In short, all reviewers felt that the experimental content of the work was incremental and that the model-based definition of defects was not sufficiently validated.

Each reviewer proposed additional analyses to validate different aspects of the defect definition or the interpretation of the experimental measurements. The authors chose not to implement most of these suggestions. Instead, they appropriately addressed the reviewers' concerns directly, rendering the suggested analyses unnecessary. Accordingly, the authors revised the manuscript and significantly strengthen it.

I believe the paper, in its current form, should be accepted for publication.

We thank Referee 3 for the overall comments and for recommending our work for publication.

Minor comment: The response to reviewer-1's question, "Why choose $\omega=50$ value?" is not entirely convincing. Based on the authors' reply, I believe they should have provided a clearer explanation of the parameters in the system that determine this value.

We are sorry that our previous reply about the cutoff $\omega = 50$ was not very clear. The value of the cutoff is not chosen using any physical arguments but rather by looking at the convergence of the numerical procedure to evaluate the softness parameter. In Suppl. Fig.5 we show the softness parameter using different frequency cutoffs. We observe that the results converge approximately around $\omega_{cut} = 50$, while they are qualitatively different if the cutoff is taken to be a smaller value of frequency. Hence, we decided that $\omega_{cut} = 50$ is an optimal cutoff value that provides a stable and robust estimate of the softness. We added an explanation in the supplementary file to clarify this point further.

Referee 4

As I understand it, the present work presented in this paper is a hybrid of experiment and numerics: an experimental colloidal glasses actual configuration is then simulated to compute a vibration spectrum. The idea is to experimentally confirm that some elements of the soft spots in the colloidal glass are correlated to the "defect" distribution of the vibration eigenmodes of lowest frequency. The key paper to compare to are simulations in Ref. 38, which show a correlation between plastic events and topological defects.

The paper has obviously been reviewed by many experts already. I don't want to torture the authors too much on what is clearly an honest and reasonable work. I do have questions though that I would be interested seeing addressed.

We thank the Referee 4 for reviewing our work and for the positive assessment.

One key difference between the figures of Ref. 38 and those of the present work that are bothering me are that the experimental eigenmodes seem to have large swaths of small magnitude vectors away from the defects (how small is not clear though so perhaps I am seeing an artifact of the graphics). That is, the regions between defects appear to be much less ordered here than in Ref. 38, and many of the defects identified appear in these low order regions (I am abusing the analogy to stat mech here a little). It makes intuitive sense that anti vortices would signify places that rearrangements might happen but surely the magnitude of the vibration around that defect must be important there? Might the authors at least comment on the apparent differences in how the eigenmodes actually look?

We appreciate the reviewer's insightful observation regarding the visual differences between the eigenvector fields in our study and those in Ref. 38 (Wu et al., Nat. Commun., 2023). Indeed, as the reviewer notes, the regions between defects appear to be much less ordered in our experimental system than in Ref. 38. This is a visual artifact that arises because the simulated eigenvector field in Ref. 38 exhibits less variation in magnitude at low frequencies compared to our experimental system.

In our case, the magnitude of the eigenvector field is broader, and we utilize a color bar to represent the phase angle of the eigenfield (θ). The regions between defects in our system are also well-ordered, which can be interpreted through the similar color in those regions. To clarify this, we have now added Supplementary Fig. 5d-f, where the normalized field is presented using Schlieren patterns, showing the ordered structure between defects at low frequencies more clearly. We notice that the broader eigenvector magnitude in our system, compared to the 2D simulated glass of Wu et al., can be attributed to several factors and differences between our experimental system and the simulated system of Wu et al. These differences include, in particular: thermal fluctuations in the experimental system (whereas the simulated system is athermal), and the interparticle interactions in our experimental system, which are quite different from the Lennard-Jones type potential used in the simulations of Wu et al.

It is also important to note that when identifying the winding number or topological charge, only the phase angle of the field is considered, with the magnitude of the eigenfield being disregarded. While vibrations around defects may indeed hold some significance, this aspect does not directly arise from the topological charge definition.

There are also other ways (maybe better ways) of visualizing topological defects beyond these discrete winding numbers (see Hoffmann, Karl B., and Ivo F. Sbalzarini. "Robust-

ness of topological defects in discrete domains." *Physical Review E* 103, no. 1 (2021): 012602 on identifying defects in discrete systems). I think it might be useful to at least check that your identification of the defects are robust to other methods of identifying topological defects. A method I especially like is to look at the curves that are spanned by all the arrows pointing near a particular direction (maybe colored by intensity). This would be like looking at a Schlieren texture in a 2D nematic liquid crystal. I believe this is called the Pontryagin-Thom construction. These lines terminate on or near the defect cores (ideally) because defects are characterized by the vector order pointing in all directions as you traverse the core. See, for example, Alexander, Gareth P. "Topology in liquid crystal phases." *The role of topology in materials* (2018): 229-257.

We thank the reviewer for this thoughtful suggestion. We agree that topological defects can be identified using various methods, and we acknowledge the robustness of the approach outlined in Hoffmann and Sbalzarini (*Physical Review E*, **103**, 012602, 2021), which we have now cited in the revised manuscript.

To ensure the robustness of our findings, we have performed a defect analysis by varying the interpolation grid lengths (r_c). We found that the defect structure and behavior remain unchanged for slight variations in r_c about the average interparticle distance measured experimentally. Obviously, the grid size cannot be much smaller than the average interparticle distance, or otherwise many cells will remain empty; conversely, it cannot be much larger than the average interparticle distance, or otherwise there will be cells with multiple occupancy. We have included a new section "**Robustness of topological defects**" in the supplementary information to address this matter. In response to the reviewer's suggestion, we have also visualized the field structure using Schlieren patterns, where color regions merge or emanate from the defect cores (see Supplementary Fig. 5d-f). Both the manuscript and supplementary material have been revised accordingly.